# Compound Fault Diagnosis of Planetary Gearbox Based on Improved LTSS-BoW Model and Capsule Network

**DOI:** 10.3390/s24030940

**Published:** 2024-01-31

**Authors:** Guoyan Li, Liyu He, Yulin Ren, Xiong Li, Jingbin Zhang, Runjun Liu

**Affiliations:** 1Key Laboratory of Advance Transducers and Intelligent Control System, Ministry of Education, Taiyuan University of Technology, Taiyuan 030024, China; 15536645231@163.com (L.H.); renyulin991003@163.com (Y.R.); 19873375060@163.com (X.L.); zhangjingbin04@163.com (J.Z.); 13277234451@163.com (R.L.); 2School of Mechanical and Electrical Engineering, University of Electronic Science and Technology of China, Chengdu 611731, China

**Keywords:** planetary gearbox, compound fault diagnosis, capsule network, local temporal self-similarity, bag of words model

## Abstract

The identification of compound fault components of a planetary gearbox is especially important for keeping the mechanical equipment working safely. However, the recognition performance of existing deep learning-based methods is limited by insufficient compound fault samples and single label classification principles. To solve the issue, a capsule neural network with an improved feature extractor, named LTSS-BoW-CapsNet, is proposed for the intelligent recognition of compound fault components. Firstly, a feature extractor is constructed to extract fault feature vectors from raw signals, which is based on local temporal self-similarity coupled with bag-of-words models (LTSS-BoW). Then, a multi-label classifier based on a capsule network (CapsNet) is designed, in which the dynamic routing algorithm and average threshold are adopted. The effectiveness of the proposed LTSS-BoW-CapsNet method is validated by processing three compound fault diagnosis tasks. The experimental results demonstrate that our method can via decoupling effectively identify the multi-fault components of different compound fault patterns. The testing accuracy is more than 97%, which is better than the other four traditional classification models.

## 1. Introduction

### 1.1. Literature Review

Planetary gearboxes play an important role in mechanical equipment such as wind turbine, helicopter and construction machinery, which generally work under time-varying load conditions. The key parts of a planetary gearbox are prone to multiple structural damages such as wear, broke, pitting and crack, etc. due to the influence of long-term alternating stresses. The service performance of a planetary gearbox further endangers the operation safety of the entire mechanical equipment. Therefore, it is significant to diagnose the potential faults of a planetary gearbox [1,2].

The internal components of a planetary gearbox are varied and generally work together with a complex coupling relationship. The fault characteristics are coupled in that the failure of several components may simultaneously occur to different degrees. Moreover, the fault features could be impacted by multi-source excitations such as random impacts, time-varying load, strong noise, multi-interface attenuation and so on. As a result, it is very difficult to identify the compound fault of a planetary gearbox [3,4].

A series of studies have been carried out for the signal processing methods of a compound fault diagnosis. Since the fault signal is highly unstable with complex frequency components, time–frequency methods such as the wavelet transform, ensemble empirical mode decomposition (EEMD), symplectic geometric mode decomposition (SGMD), local mean decomposition (LMD), local characteristic-scale decomposition (LCD), variational mode decomposition (VMD), etc. are mainly used. Teng et al. [5] proposed a modulation model based on wavelet transform, which provided an effective tool for wind power gearbox compound fault diagnosis; Zhao et al. [6] used the EEMD and feature fusion methods to diagnose the composite fault of rolling bearing; Pan et al. [7] proposed an SGMD signal decomposition algorithm to decompose the compound fault signals of rotating machinery; Huang et al. [8] combined recursive least squares (RLS) with LMD to diagnose the early fault of bearings; Wang et al. [9] proposed an improved LCD method to extract the early fault characteristics of bearings; Zhang et al. [10] combined VMD with adaptive maximum correlated kurtosis deconvolution (AMCKD) to detect the wind turbine rolling bearing faults. Above all, these methods mainly focus on the improvement of the signal decomposition ability. However, the subsequent compound fault separation and identification rely heavily on expert experience and knowledge, resulting in low recognition accuracy.

Deep learning (DL) has been used increasingly in the intelligent diagnosis of mechanical equipment. The typical DL-based methods for the intelligent diagnosis of a planetary gearbox include: Deep Belief Network (DBN) [11,12], Generative Adversarial Network (GAN) [13], Convolutional Neural Network (CNN) [14,15,16], Long Short-Term Memory (LSTM) [17], etc. In terms of compound fault diagnosis, Shao et al. [12] combined adaptive DBN and CNN to diagnose the multiple faults of rolling bearings; Zhao et al. [13] proposed a GAN model to improve the diagnosis performance under data imbalance conditions; Zhang et al. [14] combined fast spectral kurtosis (FSK) with multi-branch CNN for multi-fault diagnosis of wind turbine gearboxes.

However, most classification models treat compound fault as a new fault class and output single label, which cannot provide a true sense of decoupling identification of the compound fault. In fact, the compound fault is not exactly a new fault class since its fault information consists of corresponding fault characteristics of single faults. In addition, the training process of a DL-based model requires a large number of training samples. However, the fault samples are relatively rare in practice, and random combinations of different single faults can generate various compound faults. It is impractical to collect sufficient compound fault samples for classification model training. Therefore, it is necessary to propose a new intelligent diagnosis method, which is especially suitable for compound fault diagnosis, and has the following functions: (1) only the single fault samples are required for model training, and the trained model can use the fault knowledge learned from the labeled single fault samples to identify the fault components of compound fault test samples; (2) the model can predict multi-labels for test samples when making classification decisions.

The typical multi-label classification methods include binary relevance, multi-label K-NN, Convolutional Neural Networks (CNN), Recurrent Neural Networks (RNN), and Transformer structures [18,19], etc. Capsule Network (CapsNet) is a novel type of network proposed by Hinton et al. [20] in 2017. It utilizes capsule vectors rather than scalar neurons as the input and output of the network layers, which overcomes the problem that traditional networks cannot extract the spatial feature information. Meanwhile, it cancels the pooling layer to avoid the loss of valuable information, and can conduct multi-label outputs. It has been increasingly used in the fields of electroencephalography (EEG) emotion recognition [21], image and text classification [22,23], etc. In particular, it can identify and separate the overlapped objects [22], which is an important feature for the identification of compound faults.

In terms of single fault diagnosis, Liu et al. [24] proposed an improved multi-scale residual generative adversarial network (GAN) and feature enhancement-driven capsule network to solve the imbalanced fault data problem. Li et al. [25] proposed a dual convolutional–CapsNet for the fault diagnosis of a planetary gearbox under different rotation conditions. In terms of compound fault diagnosis, Liang et al. [26] integrated CapsNet with stockwell transform (ST) and data augmentation generative adversarial networks (DAGANs) to diagnose the single and simultaneous faults for a wind turbine gearbox. Xu et al. [27] developed an improved deep convolutional–CapsNet to diagnose the sun gear–planet compound faults of an RV gearbox; Huang et al. [28] adopted a convolutional–CapsNet model with a multi-label classifier to decouple the gear-bearing compound faults of automotive transmission. Then, Huang et al. [29] combined deep CapsNet and ensemble learning to improve the compound fault identification accuracy of an automotive gearbox.

To achieve accurate identification, sufficient feature information needs to be fed into the primary layer to ensure the CapsNet is working efficiently, which depends on the feature extractor of the model. However, most models use the convolutional network to extract features from raw signals; the feature extraction process is considered as a “black box”, which has the limitation in compound fault feature extraction. As a result, the classification performance of the diagnosis model has been limited. This motivates us to develop a more suitable feature extractor to directly obtain high representation feature vectors from raw signals and ultimately increase the classification ability.

To address this issue, a new feature extractor is constructed to optimize CapsNet, which combines local temporal self-similarity (LTSS) and bag-of-words (BoW) methods, named LTSS-BoW, for feature extraction. This model is improved from the temporal self-similarity method [30], which has been successfully used to recognize the image action sequences due to its advantage of cross-view structural stability. In order to reduce the feature dimension and increase the computing efficiency, a sliding window is utilized to divide the raw time-series into a local subseries. On this basis, LTSS matrices of the subseries are constructed and the gradients of LTSS matrices are calculated. Then, the multi-dimensional LTSS feature vectors are obtained by moving the sliding window with a fixed step size to traverse the entire sample signal. The LTSS feature extraction leads to much data redundance, and thus brings a large computation burden. Therefore, BoW is utilized to further improve computing efficiency, which has the advantages of strong anti-noise ability and good robustness [31,32]. Finally, the histogram feature vectors are treated as the inputs of the CapsNet layers.

### 1.2. Main Contributions of This Paper

The main contributions of this work are summarized as follows:(1)A novel framework, named LTSS-BoW-CapsNet, is proposed to intelligently identify the fault components contained in the compound fault signals of the planetary gearbox.(2)An LTSS-BoW-based feature extraction method is presented to increase the identification performance, which can be used to directly obtain high representation feature vectors from raw signals.(3)A multi-label classifier based on CapsNet is designed to predict multi-labels for compound fault classification decisions, in which the dynamic routing algorithm and average threshold are adopted.(4)Verification experiments are conducted to demonstrate the advantages of the proposed method.

### 1.3. Structure of the Rest of This Paper

The rest of this paper is organized as follows: Section 2 introduces the basic theoretical background of the LTSS-BoW feature extractor and CapsNet, and presents the overall diagnostic scheme of the proposed LTSS-BoW-CapsNet method in detail. Section 3 shows the experimental verification and comparative analysis results. The conclusions are drawn in Section 4.

## 2. Research Methodology

### 2.1. Compound Fault Description

Based on the author’s previous research works [33,34], Figure 1 exhibits the simulated vibration signals of a planetary gear set with the planet gear crack, sun gear crack and compound gear cracks, respectively. For the single crack fault cases, a series of abnormal impulses with a fixed period appear in the time-domain as the cracked tooth engages with the matching gear. As shown in Figure 1d, the compound fault-related features contain the information of two kinds of single fault-induced features. However, it is not simply the superposition of single fault-induced features. When two cracked teeth are engaged simultaneously, the two types of anomalous pulses will be coupled and form new fault features; meanwhile, the single fault-induced features are also deformed. Therefore, it is difficult to identify the fault components contained in the compound fault signal.

The main limitations of existing compound fault diagnosis models are: (1) most researchers label compound faults as a new fault class, and the compound fault samples need to be fed into the network with other single fault samples for model training [12,13,14]. Therefore, the proposed network cannot work effectively if the compound fault training samples are insufficient. However, compound faults are not exactly a new fault class since it contains the fault information of single fault components. Hence, it is possible to use the fault knowledge learned from the labeled single fault samples to identify the fault components of the compound fault; (2) the traditional neural networks generally use a softmax classifier, which only identifies the most obvious fault class [26,27]. Therefore, it cannot output multiple independent labels at the same time, so it is unable to identify via decoupling the fault components contained inside the compound fault. In addition, it is more difficult to distinguish the fault components in the condition that the single fault-related features are close to each other.

### 2.2. Overall Framework of the Proposed Method

In order to intelligently identify the fault components contained in the compound fault of the planetary gearbox, this work proposed a novel diagnostic framework, which mainly includes two parts: the LTSS-BoW model is the feature extractor, and CapsNet makes multi-label classification decisions. The overall framework of the proposed method is shown in Figure 2. Each part is described in the following.

### 2.3. LTSS-BoW Feature Extractor

#### 2.3.1. LTSS Feature Extraction

LTSS is an optimized data representation method that utilizes local structural information of time-series. Compared with conventional statistical features, LTSS feature matrices contain more useful information including sequential characteristics and change trends.

Figure 3 shows the proposed LTSS feature extraction scheme, which includes three steps: (1) Construct the LTSS matrix from the raw signal. At first, the sliding window is utilized to divide the raw signal sample into a local subseries. On this basis, LTSS matrices of the subseries are constructed. (2) Extract the gradient feature of the LTSS matrix. The upper right triangular elements of the LTSS matrix are divided into several blocks. Then, the gradient of each block is calculated to construct the block-based descriptor. (3) Transform the signal sample to a sequence of LTSS feature vectors. The signal samples are represented as a sequence of LTSS feature vectors, and all these samples are then gathered together to form a feature space. The detailed steps are given as follows.

***Step 1:*** *Construct the LTSS matrix from the raw signal*

The raw vibration signals measured under each health condition are divided into non-overlapping and equal-length signal samples. The sliding window with a length of m=2Δt+1 is utilized to collect the data points around time point *t* from time step t−Δt to t+Δt. The dataset $y\left(t\right)$ can be denoted as:(1)$$y\left(t\right)=\left\{x\left(t-\Delta t\right),\cdots ,x\left(t\right),\cdots ,x\left(t+\Delta t\right)\right\}$$

Then, the Euclidean distance of each two data points in $y\left(t\right)$ is calculated to construct the LTSS matrix $D\left(t\right)$:(2)$$D(t)=\left[d_{ij}\right]=\left[\begin{array}{ccc} d_{11} & \ldots & d_{1m}\\ \vdots & \ddots & \vdots \\ d_{m1} & \cdots & d_{mm} \end{array}\right]$$
(3)$$d_{ij}=\left\|x\left(i\right)-x\left(j\right)\right\|$$
where $d_{ij}$ denotes the Euclidean distance between the *i*th element and the *j*th element in $y\left(t\right)$, and *m* is the length of $y\left(t\right)$, i.e., 2Δt+1.

***Step 2:*** *Extract the gradient feature of the LTSS matrix*

Since the LTSS matrix is symmetrical, only the upper right triangular elements are considered in order to save computing resources. A block-based descriptor is employed to capture the structural information hidden in the LTSS matrix. At first, the whole matrix is divided into several blocks with a size of n×n; Then, the gradient of each block is calculated to obtain column vector $b_{q}$; Finally, all these vectors are concatenated as a multi-dimensional vector $p_{t}$, which is named as the upper right triangular block-based descriptor:(4)$$p_{t}=\left[{b_{1}}^{T},{b_{2}}^{T},\cdots ,{b_{q}}^{T}\right]$$
where *T* stands for transpose; *q* is the number of blocks.

As shown in Figure 3, the detailed procedure to calculate the gradient vector $b_{q}$ is as follows. Taking a block with a size of n×n as an example, the gradient in x direction $p_{x}$ is defined as:(5)$$p_{x}=\left[l_{x1},l_{x2},\cdots ,l_{xn}\right]$$
(6)$$\left\{\begin{array}{l} l_{x1}=r_{2}-r_{1}\\ l_{xi}=\frac{r_{i+1}\;-\;r_{i-1}}{2}\\ l_{xn}=r_{n-1}-r_{n} \end{array}\right.$$
where $l_{xn}$ is the column vector of $p_{x}$; $r_{n}$ is the column vector of the block. The similar calculation can be used to get the gradient in y direction $p_{y}$.

Then, an 8-bin histogram-based gradient direction is defined as: (7)$$\left\{\begin{array}{l} angel=\arctan\left(\frac{p_{y}}{p_{x}}\right)\;\;\left(if\;p_{x},p_{y}>0\right)\\ angel=\pi +\arctan\left(\frac{p_{y}}{p_{x}}\right)\;\;\left(if\;p_{x}<0\right)\\ angel=2\pi +\arctan\left(\frac{p_{y}}{p_{x}}\right)\;\;\left(if\;p_{x}>0,p_{y}<0\right) \end{array}\right.$$

The gradient vector $b_{q}$ is formed by counting the number of elements within the range of each gradient direction.

***Step 3:*** *Transform the signal sample to a sequence of LTSS feature vectors*

A sequence of feature vectors $p_{t}$ can be obtained by moving the sliding window with a fixed step size to traverse the entire sample signal, and repeating the above steps. Then, the signal sample can be transformed to a sequence of LTSS feature vectors as:(8)$$Z=\left[p_{1}^{T},\;p_{2}^{T},\;\cdots ,\;p_{j}^{T},\;\cdots ,p_{k}^{T}\right]$$
where $p_{j}$ is the feature vector extracted from the *j*th sliding window, and *k* is the number of sliding windows.

#### 2.3.2. BoW Model

The LTSS feature extraction leads to much data redundance, and thus brings a large computation burden. The BoW model is a simplifying assumption to construct a global representation from local features, which can be used to improve computing efficiency, making it common in many fields such as natural language processing and image/video-based action recognition [31,32]. As shown in Figure 4, for the learning phase, the BoW model performs an adaptive k-means clustering algorithm to sort all original LTSS feature vectors generating a codebook. For a new feature sample, a histogram-based encoding (HBE) strategy is used to encode it based on the codebook. The statistical feature histogram is subsequently computed as the inputs of the CapsNet layers.

***Step 1:*** *Form the codebook using clustering algorithm.*

Based on Section 2.3.1, the signal samples are represented as a sequence of LTSS feature vectors $Z_{i}$, and all these samples are then gathered together to form a feature space:(9)$$T=\cup_{i=1}^{s}Z_{i}$$
where *s* is the number of samples.

In this paper, the typical *k*-means clustering algorithm is employed to automatically learn the most representative words, i.e., codewords, which are determined by the cluster centers. Further, all these codewords form a codebook with a size of K, i.e., $\left\{C_{i}\right\},\;i=1,2,\cdots ,K$, where K is the cluster number, which has great influence in clustering results.

Several approaches have been proposed to select the appropriate cluster number [35]. Among them, the Davis–Bouldin (DB) index is a promising method because of its simplicity, which is defined as the average similarity measure of each cluster with its most similar cluster, and is expressed as:(10)$$DB=\frac{1}{K}\sum_{i,j=1}^{K}\underset{i\neq j}{\max}D_{i,j}$$
where the similarity $D_{i,j}$ is the ratio of within-cluster distances to between-cluster distances (see Figure 4), and the expression is:(11)$$D_{i,j}=\frac{\bar{d_{i}}+\bar{d_{j}}}{d_{i,j}}$$
where $\bar{d_{i}}$ denotes the average distance of all points in the *i*-th cluster to the cluster center, $\bar{d_{j}}$ denotes the average distance of all points in the *j*-th cluster to the cluster center, $d_{i,j}$ denotes the Euclidean distance between the *i*-th and *j*-th cluster centers.

According to Equations (10) and (11), the smaller the DB index, the better the clustering results. So, the best cluster number corresponds to the minimum DB index.

***Step 2:*** *Encode the feature sample using histogram-based encoding (HBE) strategy*

Histogram is an accurate representation of the distribution of numerical data, which has been widely employed in image processing, quality evaluation and time-series processing [29]. Assume that a new signal sample Y has been expressed as the LTSS feature vector form, for each point of the LTSS feature vector, the Euclidean distance between it and all the codewords in the codebook are calculated, and the codeword with minimum Euclidean distance is assigned to replace this point. Thus, the LTSS feature vector is described by a series of nearest codewords, and the frequency of each codeword is gathered to construct the statistical feature histogram, i.e., $H=\left\{h_{1},h_{2},\cdots ,h_{K}\right\}$.

### 2.4. Capsule Network for Decision-Making

The framework of CapsNet is shown in Figure 5. CapsNet usually contains a primary capsule layer and digital capsule layer. Different from the traditional neural network, the main improvements of CapsNet are: (1) traditional scalar neurons are replaced by capsule vectors to further mine the spatial information of features; (2) the dynamic routing algorithm is adopted to transmit information between the primary capsule layer and digital capsule layer, which effectively reduces the loss of feature information.

The specific parameters of CapsNet include the number and dimension of the capsules. For the primary capsule layer, the number of capsules is determined by the best cluster number K of the BoW model, and the dimension of the capsules is the same as the number of sliding windows used in the LTSS model. For the digital capsule layer, the number of digital capsules is the number of classifications, and the module length of the digital capsule vector represents the classification probability. Since the digital capsules are independent of each other, it can predict multi-labels for test samples when making classification decisions; therefore, CapsNet is adopted as the fault classifier.

The entire process of the dynamic routing algorithm can be divided into four steps as follows. The margin loss function and average threshold adopted for decision-making are described in Section 2.5 and Section 2.6, respectively.

***Step 1:*** The input vectors $u_{i}$ of the primary capsule layer are the extracted feature vectors by the previous LTSS-BoW model. Each primary capsule is multiplied by an independent weight matrix to predict the high-level capsule, which can be expressed as:(12)$$u_{\langle j|i\rangle }=W_{i,j}\cdot u_{i}$$
where the subscripts i and j denote the ith primary capsule and jth digital capsule, respectively. $W_{i,j}$ is the weight matrix, and $u_{\langle j|i\rangle }$ denotes the prediction vector.***Step 2:*** The output vector $s_{j}$ is obtained by the weighted sum of all the intermediate prediction vectors $u_{\langle j|i\rangle }$, which can be expressed as:(13)$$s_{j}=\sum_{i}c_{i,j}\cdot u_{\langle j|i\rangle }$$
where $c_{i,j}$ is the coupling coefficient determined by the softmax function, which can be regarded as the connection probability that $u_{\langle j|i\rangle }$ should be coupled to $s_{j}$. The process can be expressed as:(14)$$c_{i,j}=\mathrm{softmax}\left(b_{i,j}\right)=\frac{\exp\left(b_{i,j}\right)}{\sum_{j=1}^{k}\exp\left(b_{i,j}\right)}\geq 0,\sum_{j}c_{i,j}=1$$
where *k* is the number of digital capsules. $b_{i,j}$ is the prior probability of $c_{i,j}$. In the forward propagation, $b_{i,j}$ is initialized to zero and updated by dynamic routing as Algorithm 1.***Step 3:*** The final output vector $h_{j}$ of the digital capsule layer can be obtained by the nonlinear mapping of $s_{j}$ using the squashing function. The squashing function can compress the vector modulus length within the range of $\left[0,1\right)$ without changing its orientation, which can be expressed as:(15)$$h_{j}=\mathrm{squashing}\left(s_{j}\right)=\frac{{\left\|s_{j}\right\|}^{2}}{1+{\left\|s_{j}\right\|}^{2}}\cdot \frac{s_{j}}{\left\|s_{j}\right\|}$$***Step 4:*** The dynamic routing process is executed as shown in Algorithm 1 to update $b_{i,j}$:(16)$$b_{i,j}=b_{i,j}+h_{j}\cdot u_{\langle j|i\rangle }$$
where the dot product $h_{j}\cdot u_{\langle j|i\rangle }$ is used to evaluate the similarity between the intermediate prediction vector $u_{\langle j|i\rangle }$ and the output vector $h_{j}$. The higher the similarity, the larger the values of $b_{i,j}$ and $c_{i,j}$. The optimal solution of the coupling coefficient $c_{i,j}$ can be obtained by continuous updating.

Ultimately, the final output vector hj is returned, and the modulus length of the vector represents the classification probability pjpred.
**Algorithm 1** Dynamic routing algorithm1: Enter: uj|i2: Initialization parameters: bi,j0
3: Set the number of iterations T4: For r=1 to T  do5: ci,jr=softmaxbi,jr−1
6: sjr=∑iuj|ici,j
7: hjr=squashingsjr
8: bi,jr=bi,jr−1+hjr⋅uj|iReturn hjAmong them, softmax:  ci,j=expbi,j∑j=1kexpbi,jsquashingsjr=sjr21+sjr2⋅sjrsjr


### 2.5. Margin Loss Function

The margin loss function is adopted as the objective function to calculate the loss value. Compared with the cross-entropy loss function, the boundary loss function can directly measure the similarity between different classes of samples based on the Euclidean distance, which can expand the inter-class differences and reduce the intra-class differences. The expression is:(17)$$J=\sum_{j=1}^{k}L_{j}=\sum_{j=1}^{k}\left\{T_{j}\max{\left(0,m^{+}-p_{j}^{pred}\right)}^{2}+\lambda \left(1-T_{j}\right)\max{\left(0,p_{j}^{pred}-m^{-}\right)}^{2}\right\}$$In the formula, *k* is the number of fault classes. $T_{j}$ is the classification indicator function and $T_{j}=1$ represents that the input sample belongs to class *j*, otherwise $T_{j}=0$. $p_{j}^{pred}$ is the predicted probability that the input sample belongs to class *j*. $m^{+}$ denotes the expected upper bound of the predicted probability when the sample belongs to class *j*. $m^{-}$ denotes the expected lower bound of the predicted probability when the sample does not belong to class *j*. λ is the weight penalty factor.

### 2.6. Average Threshold

An adaptive average threshold φ is set to limit the number of output labels. If $p_{j}^{pred}$ is greater than the threshold φ, the *j*th class label output is 1, which means the *j*th class exists. Otherwise, the *j*th class label output is 0, which means the *j*th class does not exist. The process can be expressed as:(18)$$\varphi =average\left(p^{pred}\right)=\frac{1}{k}\sum_{j=1}^{k}p_{j}^{pred}$$
(19)$$L_{j}=1,\;\;\;if\;\;\;p_{j}^{pred}>\varphi$$
(20)$$L_{j}\in L=\left[L_{1},L_{2},\cdots L_{k}\right]$$
where $L_{j}$ is the output label of the *j*th class. *L* denotes the set of all the predicted class labels.

### 2.7. Diagnosis Process

Taking advantages of LTSS-BoW-based vibration feature extraction, coupled with CapsNet-based decision-making, a novel framework is proposed to diagnose the compound fault of a planetary gearbox. To summarize, the detailed steps are given below and shown in Figure 6.

(1)Collect the vibration signals of the planetary gearbox in different health states, divide the raw signal into equal-length signal samples and normalize the data samples.(2)Divide the dataset into a training dataset and a test dataset. Note that the training dataset only contains the normal and single fault samples. The test dataset is composed of compound fault samples.(3)Design the LTSS-BoW feature extractor and convert all the samples into feature matrices.(4)Train the CapsNet model based on the training dataset. The trained model is used to identify the fault components of the test samples and output the predicted probability of each fault class.(5)Compare the predicted probability of each fault class with the average threshold for class label output.

## 3. Experimental Verification

To evaluate the effectiveness of the proposed LTSS-BoW-CapsNet diagnosis method, a series of experiments were conducted on our planetary gearbox test rig. The experimental results are analyzed in three aspects: (1) demonstrate the multi-label output results of CapsNet; (2) compare the diagnosis results of our proposed method with other methods; (3) perform feature visualization to further evaluate the feature learning ability of the proposed method on the compound fault diagnosis tasks.

### 3.1. Experimental Setup and Data Description

As shown in Figure 7, the test rig for a planetary gearbox consists of the drive motor, planetary reducer, magnetic powder brake, three-axis acceleration sensor installed on the gearbox and the Dewetron acquisition system. As shown in Figure 8, four kinds of single fault patterns were separately seeded on the planetary gearbox, which are sun gear tooth crack, planet tooth crack, planet tooth surface pitting and ring gear tooth crack, denoted as SC, PC, PP and RC, respectively. Three kinds of compound fault patterns were simulated in the experiments, which are SC–PC, SC–PP and SC–RC, respectively.

In the experiment, the sun gear is the input component, and the carrier is the output. The rotation speed is set to 1200 rpm, and the load torque is 5 N·m. Setting the sampling frequency to 10,240 Hz, the vibration signal for each normal or faulty pattern is collected with a sampling time of 30 s. The raw signal is divided into non-overlapping signal samples and each sample has 2048 data points.

In order to reduce the impact of raw data on the diagnostic model, the normalization regularization method is adopted to normalize the raw data to between 0 and 1, and the corresponding formula is as follows:(21)$$M_{i}=\frac{N_{i}-\mathrm{average}\left(N_{i}\right)}{\max\left|N_{i}-\mathrm{average}\left(N_{i}\right)\right|}$$

Figure 9 shows the normalized time-domain signal for each fault pattern. It can be observed that the gear fault can induce impacts with a fixed period, i.e., $t_{s}$, $t_{p}$, $t_{r}$, in the time-domain signal. Compared with the single fault patterns, the coupling effect between multiple faults makes the vibration characteristics more complicated in compound fault cases. It is worth noting that new fault features occur due to the coupling effects, i.e., $t_{sp}$ and $t_{sr}$; meanwhile, the single fault-induced features are also deformed. Therefore, it is difficult to manually identify the compound fault components from the raw signal.

For the proposed diagnostic approach based on LTSS-BoW and CapsNet, three compound fault diagnosis tasks shown in Table 1 are set up. The normal and single fault samples are used for training based on the 5-fold cross-check method. After the model training process, the compound fault samples are used for testing. The trained model needs to predict multiple fault labels for the compound fault test samples based on the knowledge learned from the single fault samples.

### 3.2. Parameter Setting

#### 3.2.1. Parameters of LTSS-BoW Model

The length of sliding window m=2Δt+1 and cluster center number *K* have a great influence in the calculation efficiency and accuracy. Considering the calculation complexity of the LTSS matrix, the parameter Δt is set to take value from the range $\left[4,16\right]$ with a step size of 3. K is adaptively determined by DB index as described in Section 2.3.2. The smaller the DB index, the better the clustering results, so the best cluster number is set to 125 as shown in Figure 10. For each sample, the output matrix size is 125×5 after extracting basic features through the LTSS-BoW model.

#### 3.2.2. Parameters of CapsNet

The extracted features are fed into the primary capsule layer as inputs. The number of digital capsules is determined by the number of categories to be classified. During the training process, the Adam optimizer with the initial learning rate of 0.001 is adopted to update the parameters. The iteration of dynamic routing r is set to 3. The batch size is set to 10. The margin loss function is adopted to calculate the loss value. The structural parameters of the network in this paper are greatly reduced, which is more conducive to improve the training speed. The specific parameters used for the LTSS-BoW model and CapsNet are summarized in Table 2.

Our approach is based on the Pytorch framework and trained on an NVIDIA RTX3070 GPU.

### 3.3. Diagnosis Results

#### 3.3.1. The Predicted Probability for Multi-Label Output

For compound fault diagnosis task 1, the predicted probability values for each pattern are listed in Table 3 and shown in Figure 11. The LTSS-BoW-CapsNet model is trained based on the signal samples of normal, SC and PC patterns. Then, the trained model is tested based on the signal samples of an SC–PC compound fault pattern. In our experiments, each task was performed independently ten times to eliminate the influence of randomness. For all tests, the predicted probability values for the existence of SC and PC fault patterns are significantly higher than the average threshold, while the predicted probability values for normal patterns are far below the threshold. Therefore, the class labels of SC and PC fault patterns are equal to 1. Thus, the model accurately identifies the fault components of the SC–PC compound fault pattern and can output two single labels simultaneously. A similar procedure can be used to analyze the multi-label output results in task 2 and 3.

#### 3.3.2. Comparative Analysis

To validate the effectiveness of the proposed LTSS-BoW-CapsNet method, four models were selected to compare the diagnosis performance. The comparison models include an SVM-based model, a kNN-based model, a CNN-based model and a CNN-CapsNet model, which are briefly described below:(i).SVM-based and kNN-based models. To compare the effect of a classifier, two widely used classifiers SVM and kNN are used for making classification decisions. These two methods extract features based on the same LTSS-BoW model.(ii).CNN-based model. CNN is a typical neural network with convolution and pooling operations. The classical LetNet5 model is used here for comparison.(iii).CNN-CapsNet model [26]. This method uses a convolution network as a feature extractor, and a capsule network as the classifier. The parameter settings are described in Ref. [26].

In our experiments, each diagnosis task was performed independently ten times to obtain the average accuracy. Table 4 lists the accuracies of the four models on the compound fault diagnosis tasks. The results show that the SVM-based, kNN-based and CNN-based models failed in all three tasks due to the limitation of the classification principle. The CNN-CapsNet model only identifies the fault components of SC–RC and failed in the tasks SC–PC and SC–PP. The proposed LTSS-BoW-CapsNet method performed well in all tasks with an accuracy of more than 97%. This demonstrates that the proposed method can identify via decoupling the fault components and have better stability for different types of compound fault diagnosis.

To further visually analyze the diagnosis results, the confusion matrices and the label outputs of four diagnosis methods (LTSS-BoW-SVM, CNN, CNN-CapsNet and the proposed LTSS-BoW-CapsNet) for three diagnosis tasks are displayed in Figure 12 and Figure 13, respectively.

It can be clearly seen that the LTSS-BoW-SVM and CNN models only output single class labels. The reason is that the traditional classifiers identify the most obvious features and output the most likely single fault label for compound fault diagnosis task. Therefore, the traditional classifiers cannot output multiple independent labels at the same time; it is unable to identify via decoupling the fault components contained in the compound fault.

CNN-CapsNet identifies the SC and RC fault components contained in the SC–RC compound fault, while wrongly identifying the SC–PC and SC–PP compound faults as an SC single fault. The reason could be the fault features of SC are more obvious than PC and PP faults in the compound fault signals. Therefore, the CNN has the limitation in compound fault feature extraction, especially in the case that one fault component has greater influence than the other one.

The proposed LTSS-BoW-CapsNet model successfully identifies the fault components contained in the compound faults in three tasks, which indicates that the LTSS-BoW model has better feature extraction ability. Moreover, the CapsNet model can output multi-labels due to its unique classification principle. Above all, the proposed model has significant advantages in compound fault diagnosis.

Additionally, a t-SNE visual diagram is used to downscale the deep feature embedding and obtain the feature distribution of the CNN-CapsNet and LTSS-BoW-CapsNet models. As shown in Figure 14, comparing the feature results extracted by the high-level capsule layer in task 1, it can be seen that the class spacing between the SC–PC pattern and SC pattern is much closer than the class spacing with the PC pattern for the CNN-CapsNet method, which makes it easy to mistakenly identify the SC–PC class as SC. However, the class spacing distribution between the SC–PC pattern and two single fault patterns is uniform and clear for the LTSS-BoW-CapsNet method, so the fault components contained in the compound fault can be effectively identified. It indicates that the use of LTSS-BoW can enhance the ability of the network model to extract coupling features.

## 4. Conclusions

In this paper, a novel LTSS-BoW-CapsNet framework is proposed to diagnose the compound fault of a planetary gearbox. An improved LTSS-BoW feature extractor is constructed to extract fault feature vectors, which has the advantages of high feature extraction efficiency and strong robustness. Then, a multi-label classifier based on CapsNet is designed. The dynamic routing algorithm and average threshold are adopted to predict multi-labels for compound fault components recognition.

The effectiveness of the proposed LTSS-BoW-CapsNet method is evaluated by processing three compound fault diagnosis tasks. The experimental results demonstrate that our proposed approach can effectively identify via decoupling the multi-fault components contained in the compound fault signals of planetary gearbox. The testing accuracy is more than 97%, which is better than the other four traditional classification models. The trained model can only use the fault knowledge learned from the labeled single fault training samples to identify the fault components of compound fault test samples. Therefore, it can solve the problem that the compound fault samples are insufficient in practice.

This research only realized the diagnosis of compound faults containing two types of faults. However, the compound fault of a planetary gearbox could be more complex in practice. Therefore, in future work, the proposed method would be improved by using multi-channel signal fusion and feature fusion, so that it can identify more fault components via decoupling and achieve better identification performance.

## Figures and Tables

**Figure 1 sensors-24-00940-f001:**
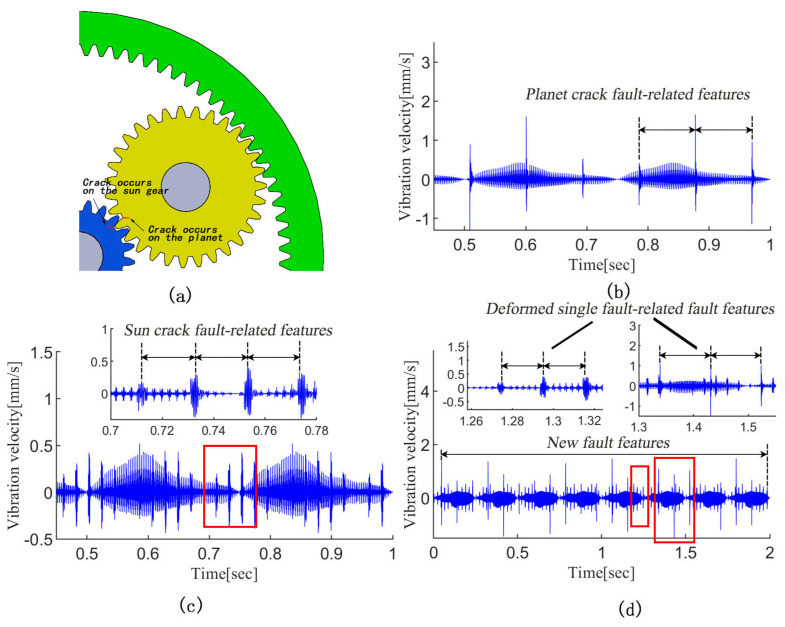
The simulated vibration signals of a planetary gear set with gear cracks. (**a**) Planetary gear set; (**b**) planet gear crack; (**c**) sun gear crack; (**d**) compound gear cracks.

**Figure 2 sensors-24-00940-f002:**
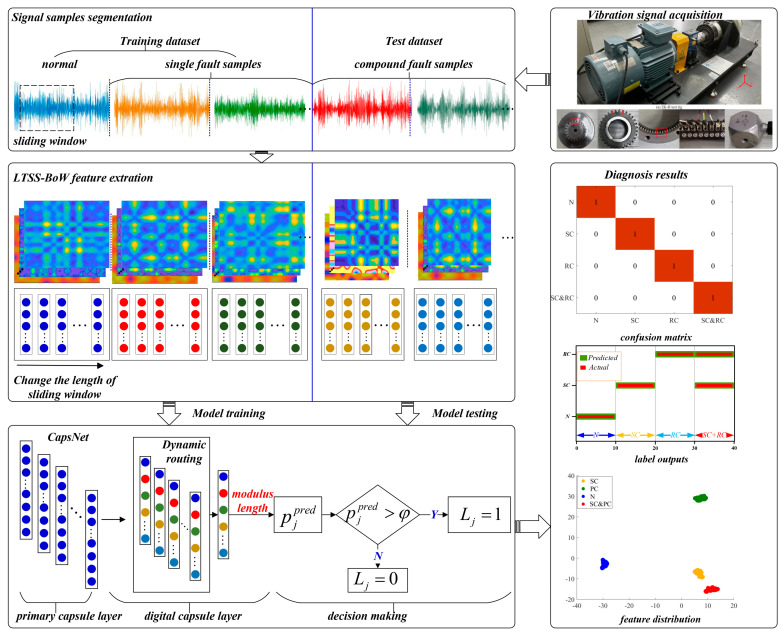
Overall framework of the proposed method.

**Figure 3 sensors-24-00940-f003:**
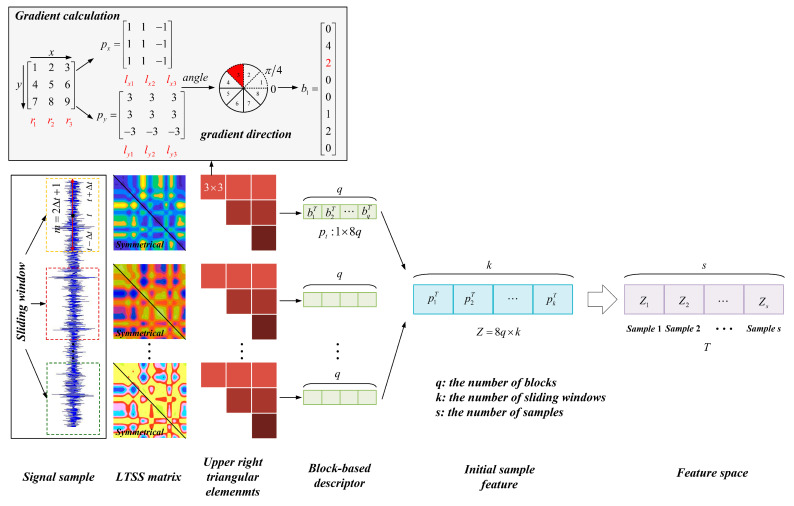
The flowchart of LTSS model.

**Figure 4 sensors-24-00940-f004:**
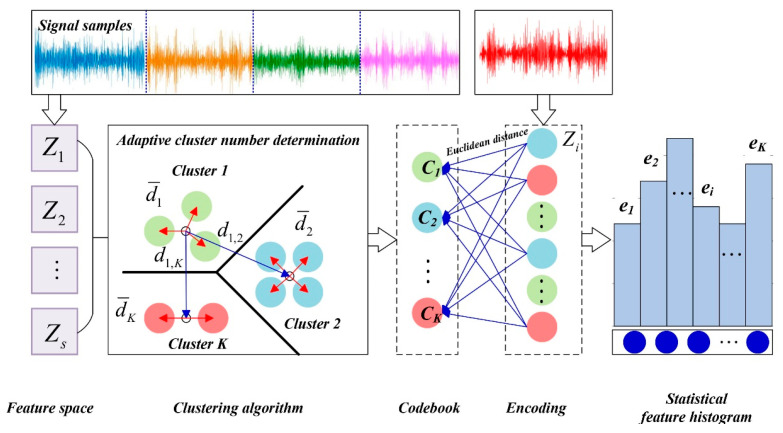
The flowchart of BoW model.

**Figure 5 sensors-24-00940-f005:**
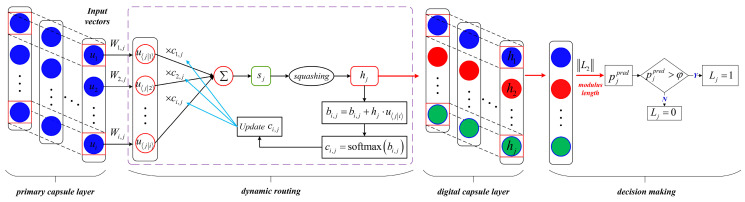
The framework of CapsNet.

**Figure 6 sensors-24-00940-f006:**
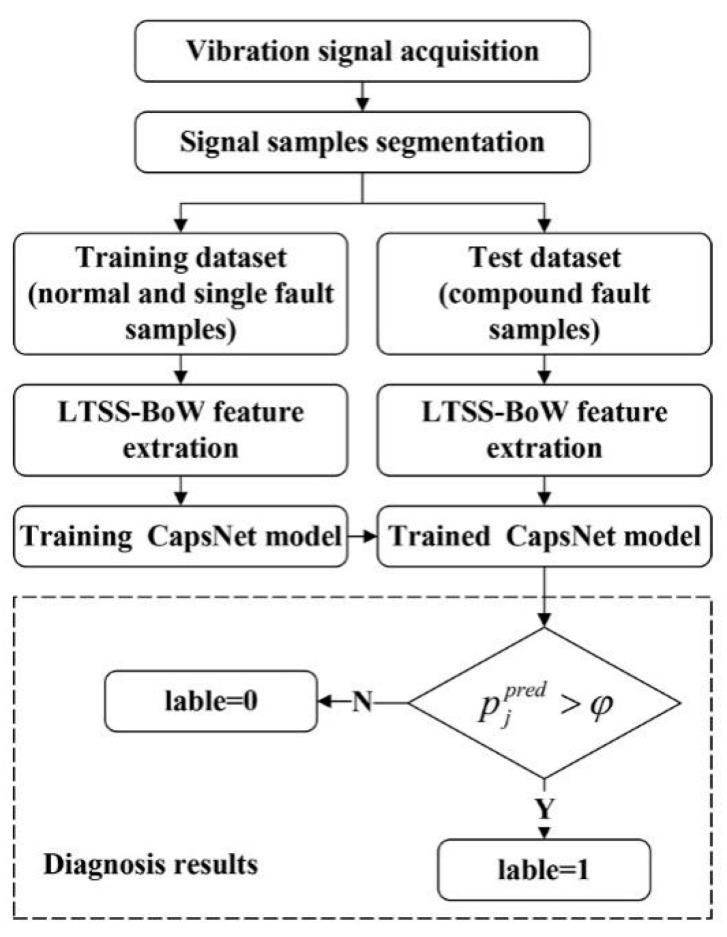
The diagnosis flowchart of proposed method.

**Figure 7 sensors-24-00940-f007:**
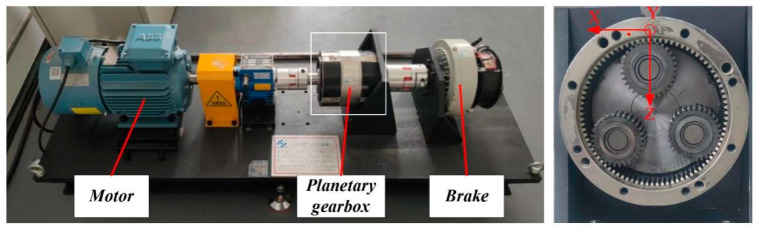
Planetary gearbox test rig.

**Figure 8 sensors-24-00940-f008:**
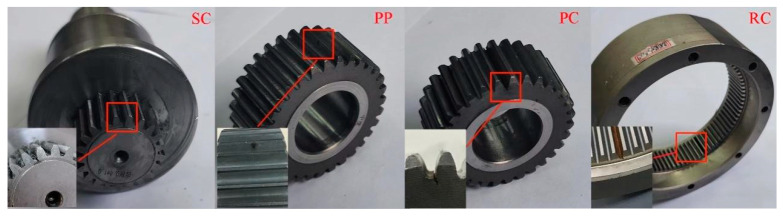
Gear faults.

**Figure 9 sensors-24-00940-f009:**
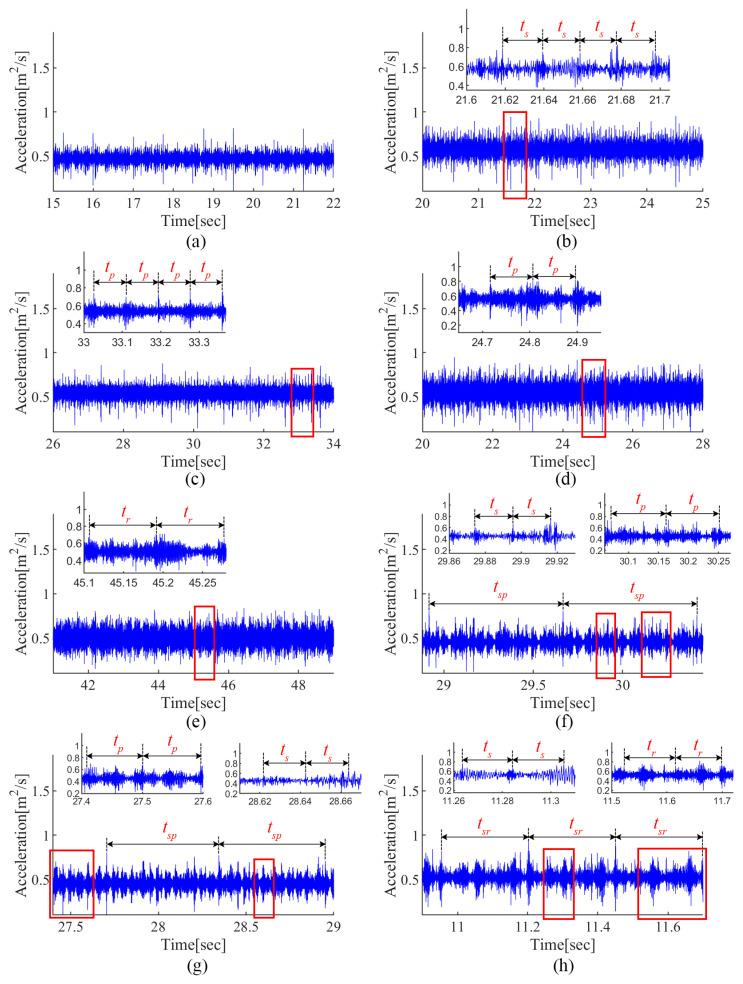
Normalized time-domain signal for each fault pattern (**a**) N, (**b**) SC, (**c**) PC, (**d**) PP, (**e**) RC, (**f**) SC–PC, (**g**) SC–PP and (**h**) SC–RC.

**Figure 10 sensors-24-00940-f010:**
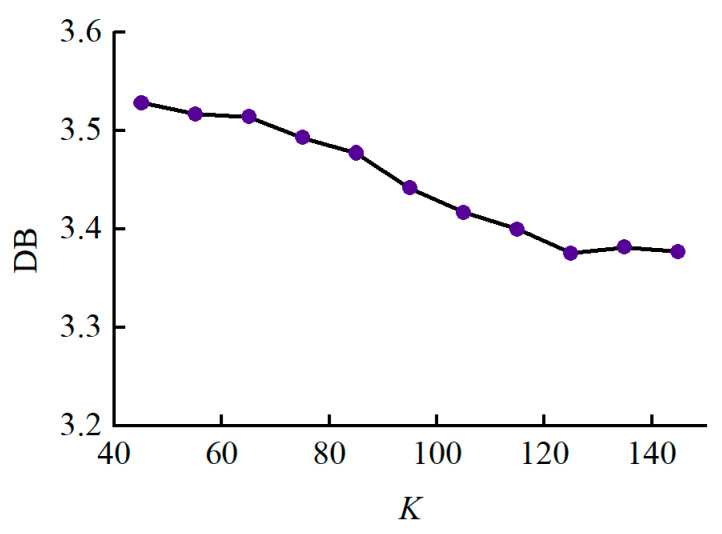
Trend of DB index.

**Figure 11 sensors-24-00940-f011:**
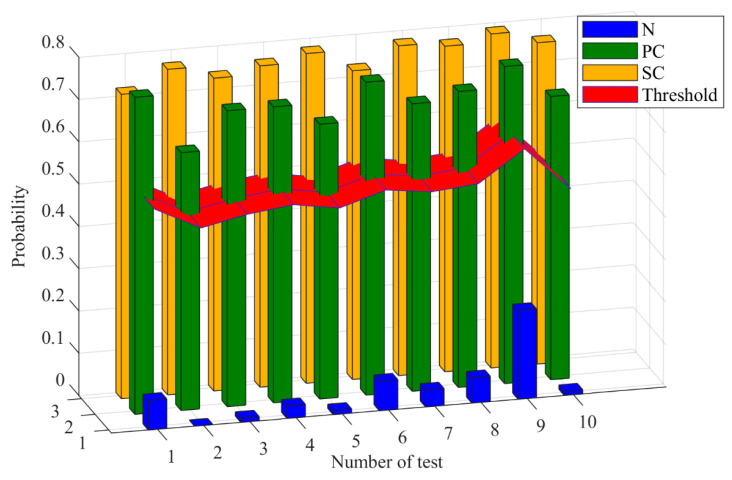
The predicted probability values for each pattern in task 1.

**Figure 12 sensors-24-00940-f012:**
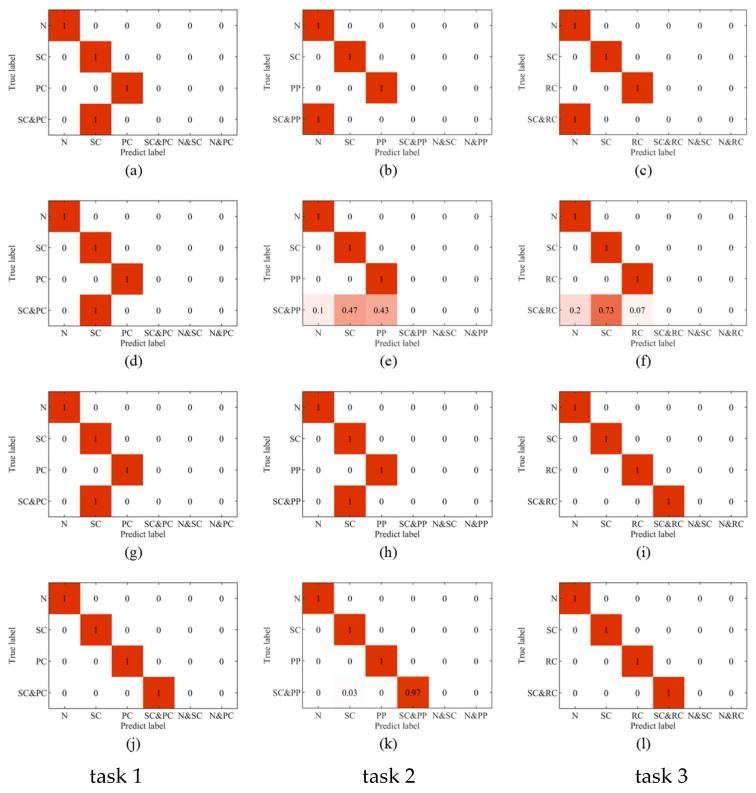
Confusion matrices of four diagnosis methods for three compound fault diagnosis tasks (**a**–**c**) LTSS-BoW-SVM model; (**d**–**f**) CNN model; (**g**–**i**) CNN-CapsNet model; (**j**–**l**) LTSS-BoW-CapsNet model.

**Figure 13 sensors-24-00940-f013:**
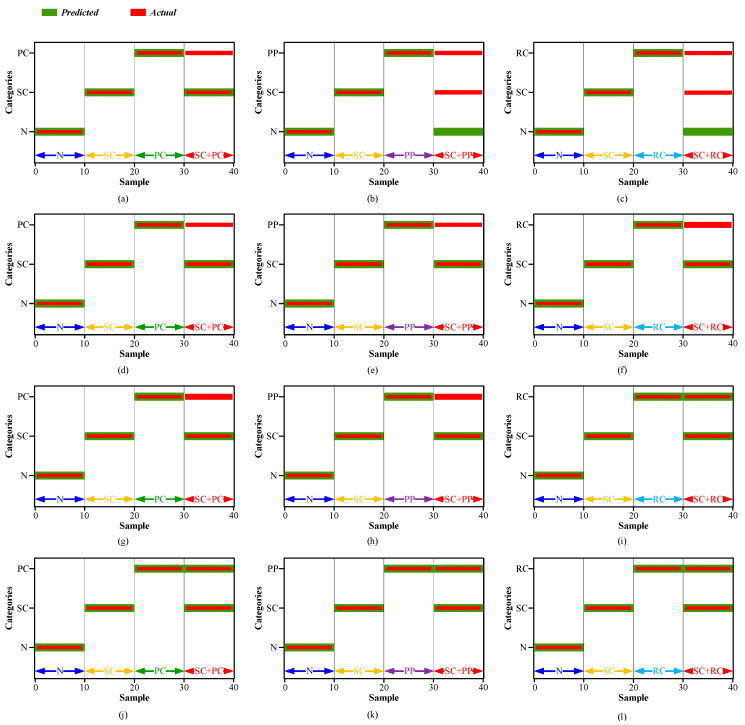
Label outputs of four diagnosis methods for three compound fault diagnosis tasks (**a**–**c**) LTSS-BoW-SVM model; (**d**–**f**) CNN model; (**g**–**i**) CNN-CapsNet model; (**j**–**l**) LTSS-BoW-CapsNet model.

**Figure 14 sensors-24-00940-f014:**
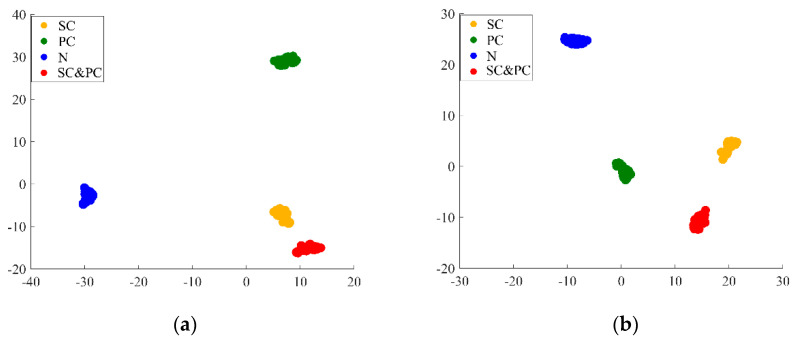
t-SNE visual diagrams in task 1 (**a**) CNN-CapsNet, (**b**) LTSS-BoW-CapsNet.

**Table 1 sensors-24-00940-t001:** Compound fault diagnosis tasks.

Task	Test Dataset	Training Dataset	Training Samples	Test Samples
1	SC–PC	N, SC, PC	100	10
2	SC–PP	N, SC, PP	100	10
3	SC–RC	N, SC, RC	100	10

**Table 2 sensors-24-00940-t002:** Model parameters.

Layer	Parameters	Value
LTSS-BoW	Δt	4, 7, 10, 13, 16
	The cluster center number *K*	125
CapsNet	The number of primary capsules	125
	The dimension of primary capsules	5
	The number of digital capsules	3
	The dimension of digital capsules	10
	The iteration of dynamic routing r	3

**Table 3 sensors-24-00940-t003:** The predicted probability for each pattern in task 1.

Number of Tests	Predicted Probability			Average Threshold
	SC	PC	N	
1	0.72	0.75	0.07	0.5133
2	0.77	0.61	0	0.46
3	0.74	0.7	0.01	0.4833
4	0.76	0.7	0.03	0.4967
5	0.78	0.65	0.01	0.48
6	0.73	0.74	0.07	0.5133
7	0.78	0.68	0.04	0.5
8	0.77	0.7	0.06	0.51
9	0.79	0.75	0.21	0.5833
10	0.76	0.67	0.01	0.48

**Table 4 sensors-24-00940-t004:** Diagnosis accuracies of all the models.

Method	Task 1	Task 2	Task 3
LTSS-BoW-SVM	0%	0%	0%
LTSS-BoW-kNN	0%	0%	0%
CNN	0%	0%	0%
CNN-CapsNet	0%	0%	100%
Our proposed method	100%	97%	100%

## Data Availability

Data are contained within the article.

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
