# Peer review of "Compound Fault Diagnosis of Planetary Gearbox Based on Improved LTSS-BoW Model and Capsule Network"

_sensors, 2024, doi:10.3390/s24030940_

Round 1

Reviewer 1 Report

Comments and Suggestions for Authors

This manuscript proposed a novel diagnostic framework, named LTSS-BoW-CapsNet, for the identification of compound fault components of planetary gearbox.The effectiveness of the proposed method is validated by the experiment results, which indicate that the proposed approach is better suited for detecting multi-fault, and has better stability and diagnostic accuracy than traditional methods. The topic is interesting and the contributions have been identified clearly. This research work is well structured containing interesting results in general, and I recommend a minor revision. A few issues are list below.

1.        In the introduction, the typical multi-label classification methods are not mentioned, which needs to be supplemented in the manuscript for better understanding.

2.        In Fig.1, new fault features appear in the simulated vibration signals of a planetary gear set with compound gear cracks, what causes new fault features? Can it be explained more?

3.        What is the advantage of using adaptive average threshold rather than the fixed threshold?

4.        The paper was structured well in general but should be proofed to avoid typos and grammar mistakes.

Reviewer 2 Report

Comments and Suggestions for Authors

The paper proposes a method for compound fault diagnosis of planetary gearboxes based on an improved local temporal self-similarity coupled with bag of words models (LTSS-BoW) and a capsule network (CapsNet). The LTSS-BoW feature extractor is used to extract fault feature vectors from raw signals, and the CapsNet is designed as a multi-label classifier with a dynamic routing algorithm and average threshold. The paper’s objective has been reached systematically; however, there are areas where the authors could improve before being accepted, such as:.

1-     In the abstract, please provide specific results or metrics to support the claim that the proposed method is better suited for detecting multi-fault components and has better stability and accuracy.

2-     The title of subsection 1.2 is not indicative.

3-     The presented contributions in subsection 1.2 need to be rewritten to make them clearer. Also, I think point 4 is not a contribution; please check.

4-     In section 2, clearer definitions of key terms, such as "local temporal self-similarity" and "bag of words models", are needed to help readers who are not familiar with these concepts.

5-     The CapsNet architecture is briefly described, but there is not enough detail provided on how it was implemented or how it works. It would be helpful to provide more details on the CapsNet architecture and the specific parameters used in the implementation.

6-     Figure 3 has to be adequately discussed.

7-     Section 3 does not provide enough details on the experimental setup. It is not clear how the data was preprocessed, what parameters were used for the LTSS-BoW model and CapsNet, and how the performance of the models was evaluated.

8-     Also, the section does not discuss the limitations of the proposed method. It would be helpful to know what the limitations are and how they could be addressed in future work.

9-     Please refer to the software and hardware that have been used to develop the proposed algorithm.

10-  The conclusion section should provide a clear and concise summary of the main findings of the study. It should highlight the key contributions and results of the research.

Comments on the Quality of English Language

Minor editing of English language required

Round 2

Reviewer 2 Report

Comments and Suggestions for Authors

Thanks for accurately considering all the given comments.